# Health-Related Quality of Life due to malaria in the Brazilian Amazon using EQ-5D-3L

**Mônica Viegas Andrade[1,2], Kenya Valeria Micaela de Souza Noronha[1,2], Gilvan Ramalho Guedes[1], Nayara Abreu Julião[1,2], Lucas Resende de Carvalho[1], Aline de Souza[1], Valéria Andrade Silva[1], Andre Soares Motta-Santos[1,2], Henrique Bracarense[1], Cássio Peterka[3], Marcia C. Castro⬡[4]***

1 Center for Development and Regional Planning, Universidade Federal de Minas Gerais, Belo Horizonte, Minas Gerais, Brazil, 2 Center for Health Technology Assessment, Universidade Federal de Minas Gerais, Belo Horizonte, Minas Gerais, Brazil, 3 Ministry of Health, Brasília, Distrito Federal, Brazil, 4 Department of Global Health and Population, Harvard T.H. Chan School of Public Health, Boston, Massachusetts, United States of America

* mcastro@hsph.harvard.edu

**Data Availability Statement:** The data and code required to reproduce the results in this article are available in the GitHub repository: https://github.

## Abstract

Malaria is a mosquito-borne infectious disease caused by protozoa of the genus *Plasmodium*. Despite of the progress in malaria control in the last decades, malaria remains a major public health problem, contributing to increased morbidity and mortality in tropical and subtropical countries. Among American countries, Bolivia, Venezuela, and Brazil account for 73% of the cases. In Brazil, the majority of malaria cases is concentrated in Amazon region. This study estimated health-related quality of life (HRQoL) losses due to malaria in endemic areas of the Brazilian Amazon using the EQ-5D-3L instrument. We collected data from a convenience sample of 1,179 individuals aged 18 years or older. To measure the HRQoL loss, we matched individuals from the treatment group (with recent malaria) to those from the control group (without recent malaria) using Propensity Score Matching (PSM) and compared the difference in mean health utility between the groups. The results show a significant loss of HRQoL due to malaria. The mean utility was 0.69 and 0.83 for the treatment and control groups, respectively, representing a loss of quality of life of approximately 16.3% for individuals with recent malaria episodes. These findings underscore the importance of effective malaria prevention and treatment strategies, especially in areas where adverse socioeconomic conditions and a challenging epidemiological context exacerbate the impact of the disease. Continued investment in malaria control programs and improved access to health services are essential to mitigate the negative impact of this disease on the quality of life of affected populations.

## Author summary

Malaria is an infectious disease transmitted by mosquitoes, primarily affecting tropical and subtropical regions worldwide. Despite significant efforts to control malaria, the rates of illness and death are still important. In the Americas, Brazil, Venezuela, and Bolivia

com/mcastrolab/healthQoL_Malaria_Amazon/tree/main.

**Funding:** This study was funded by Bill & Melinda Gates Foundation and Brazilian Ministry of Health through the National Council for Scientific and Technological Development (CNPq) (Process 442842/2019-8), call CNPq/MS-SCTIE-Decit/Fundação Bill & Melinda Gates N° 23/2019 – Pesquisas de prevenção, detecção e combate à Malária. Under the terms of the Foundation's grant, we have signed a Creative Commons Attribution 4.0 Generic License to any version of the accepted manuscript that may result from this submission. M.V.A., K.V.M.S.N., and G.R.G. acknowledge CNPq for the research productivity scholarships (processes 309252/2021-0, 303459/2022-0, 317328/2021-2, respectively). K.V.M.S.N. is also supported in part by Minas Gerais Research Funding Foundation FAPEMIG (PPM-00604-17). A.S.M.S acknowledges Coordenação de Aperfeiçoamento de Pessoal de Nível Superior (CAPES) for his scholarship. The funders had no role in study design, data collection and analysis, decision to publish, or preparation of the manuscript.

**Competing interests:** The authors have declared that no competing interests exist.

account for more than 50% of the cases. This study focuses on the Brazilian Amazon, where malaria is most prevalent, accounting for 99.5% of all reported cases in the country. We aimed to understand how malaria affects individuals' health-related quality of life (HRQoL) in these endemic areas. Using the EQ-5D-3L instrument, we collected data from over 1,100 adults in endemic municipalities of the Brazilian Amazon. We compare the EQ-5D-3L utility scores for those who recently had malaria with those who did not by means of Propensity Score Matching. Our findings revealed that individuals who recently had malaria experienced a significant reduction in HRQoL. Specifically, there was a 16.3% decrease in EQ-5D-3L utility for those affected by recent malaria episodes. These results highlight the urgent need for malaria prevention and treatment programs. Effective strategies and better access to healthcare are key in reducing the disease's impact on individuals' lives, especially in regions with challenging health environments, and poor socioeconomic conditions and infrastructure.

## Introduction

Malaria is a mosquito-borne infectious disease caused by protozoa of the genus *Plasmodium*. Significant progress has been made in malaria control since the 1990s, although progress has stalled or even reversed in some countries since the late 2010s [1]. Malaria remains a major public health problem, contributing to increased morbidity and mortality in tropical and subtropical countries. In 2022, an estimated 249 million cases and 608,000 deaths from malaria occurred worldwide. Although 94% of cases occur in Africa, just three countries (Bolivia, Venezuela, and Brazil) account for 73% of the cases in the Americas [2].

The Brazilian Amazon accounts for 99.5% of all reported malaria cases in the country. Despite a 6.6% decrease in cases between 2021 and 2022, dropping from 140,488 to 131,224 cases, preliminary data from the first half of 2023 indicate an 8.7% increase over the previous year. Furthermore, the number of malaria hospitalizations and deaths in the region has been on an upward trend since 2020 [3]. Several factors contribute to the high incidence of malaria cases in the Brazilian Amazon. The region's environmental characteristics, including high temperatures, humidity, dense vegetation, and low elevation, create favorable conditions for the proliferation of vector mosquitoes [4]. The interplay between geoclimatic and socioeconomic factors (i.e., poverty levels and inadequate housing and sanitary conditions) provides minimal or no protection against mosquitoes, thereby increasing the likelihood of infection [5, 6]. It is worth noting that the border areas of the Brazilian Amazon face intense population mobility that is driven by economic opportunities, especially in mining activities, or by humanitarian crises, such as the recent one experienced in Venezuela. These migratory flows increase the risk of imported malaria cases and also contribute to the reintroduction of the disease in regions where it was previously under control [5, 7, 8]. Additionally, the lack of adequate healthcare infrastructure in indigenous lands and hard-to-reach areas (such as illegal mining areas) hinders early diagnosis and effective treatment [8].

*P. falciparum* and *P. vivax* are the most prevalent *plasmodium* species worldwide. In Brazil, the incidence of *P. vivax* infections is considerably higher than that of *P. falciparum*, representing about 85% of autochthonous cases registered in 2022 [3]. The time elapsed between mosquito bites and the first disease symptoms varies according to the parasite species: at least seven days for *P. falciparum* and 10 to 30 days for *P. vivax*. Mild symptoms are more prevalent for *P. vivax* infections and include fever, headache, and chills, while severe symptoms, such as fatigue, confusion, seizures, and difficulty breathing are more common for *P. falciparum*[2].

These symptoms can lead to important Health-Related Quality of Life (HRQoL) loss mainly due to the recurrent episodes usually associated to malaria.

Studies aiming to measure HRQoL loss due to malaria are scarce. Only a few studies conducted field research to evaluate HRQoL among patients experiencing malaria episodes [9–12]. Most used the EuroQol (EQ-5D) instrument that defines health status in five dimensions (mobility, self-care, usual activities, pain/discomfort, anxiety/depression). This instrument allows us to estimate the Quality-Adjusted Life Years (QALY) considering societal preferences for health status. All studies using EQ-5D found low levels of HRQoL among patients with malaria. However, it is not possible to infer about HRQoL losses due to the absence of control groups.

This study addresses this gap by using the EQ-5D-3L instrument to measure the loss of health-related quality of life (HRQoL) associated with malaria by comparing individuals with and without the disease. To the best of our knowledge, this is the first application of the EQ-5D-3L to individuals with malaria in the Brazilian Amazon region.

## Methods

### Ethics statement

The project was approved by the Research Ethics Committee of the Federal University of Minas Gerais (Protocol #44774921.1.0000.5149), and the use of the instrument was approved by the EuroQol Group (ID #404310). All participants signed and received a copy of a consent term authorizing their participation in the field survey.

### Study design and data collection

Field study was conducted in nine municipalities covering five states in the Brazilian Amazon region using a convenience sample (Fig 1). Municipalities were selected based on disease incidence, as measured by the 2019 Annual Parasite Index (API), and geographic accessibility conditions. Indigenous and gold mining areas were excluded. The selected municipalities are predominantly small, with the exception of Cruzeiro do Sul–with an estimated population of more than 90,000 inhabitants–and relatively underserved areas [13].

Face-to-face household interviews were conducted in April-May 2022 by an independent external research firm. Interviewers were trained and supervised by the study authors, who also conducted quality control of the data collection. The inclusion criterion for households in the study was the presence of at least one resident with a malaria episode between January 2019 and April 2022. Measuring HRQoL losses due to malaria requires a comparison between individuals with malaria (treatment group) and those without the disease (control group). Because it is difficult to find individuals with malaria during the field research, we defined individuals with a recent malaria episode, up to three months before the questionnaire administration (between January and April 2022), as the treatment group. The control group consisted of individuals without a recent malaria episode. Only one person over the age of 17 was selected from each household. Sample selection was made from a list of eligible residents compiled during the interview, according to pre-established sex and age quotas.

HRQoL was measured by the EQ-5D-3L instrument, developed by the EuroQol Group. It is widely used for measuring HRQoL in adults and validated for use in Brazil [14]. The EQ-5D-3L questionnaire comprises five dimensions (mobility, self-care, usual activities, pain/discomfort, and anxiety/depression), each with three severity levels (1—no problems, 2—some problems, and 3—severe problems). The combination of dimensions/levels generates 243 unique health states. Health states are represented by a sequence of five digits corresponding to the severity level in each dimension. For example, 11111 represents the full health state

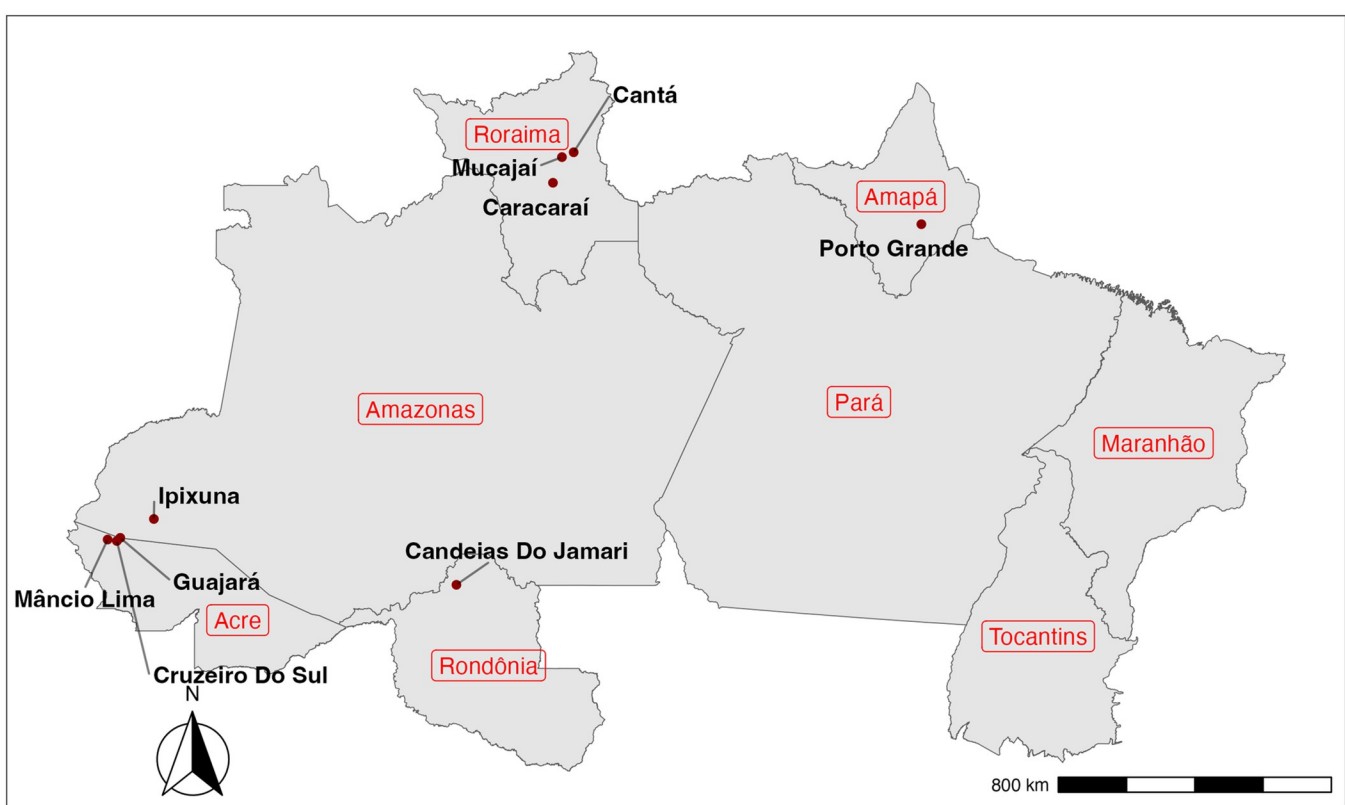

**Fig 1. Municipalities of the Brazilian Amazon selected for the study sample.** Note: Maps were created using the ggplot2 package within R environment, version 4.4.0. The base map used is derived from an openly available IBGE shape file source (https://www.ibge.gov.br/geociencias/organizacao-do-territorio/malhas-territoriais.html).

defined as having no problems in any dimension while 33333 represents the worst health state with extreme problems on all five dimensions. The EQ-5D also includes a visual analog scale (EQ-VAS), where individuals self-assess their health status on a scale ranging from 0 to 100, corresponding to the worst and best imaginable health condition, respectively. In addition to the EQ-5D instrument, interviewers also collected information on sex, age, household characteristics, self-reported general health status, and the presence of pre-existing conditions, including diabetes, hypertension, arthritis, respiratory problems, dengue, yellow fever, and COVID-19.

## Data analysis

To measure the loss of HRQoL due to malaria, individuals in the treatment group (with recent malaria) were paired with individuals in the control group (without recent malaria) using Propensity Score Matching (PSM), and the difference in mean health utility between the groups was compared. The utility associated with each EQ-5D-3L health state was obtained from the EQ-5D index values estimated by Andrade et al. [15] for the population of Minas Gerais.

PSM is used to minimize sample selection bias in observational studies. The method parametrically assigns each individual in the sample a probability of exposure to the disease–the propensity score (PS)–while controlling for observed characteristics. This creates pairs of individuals from the treatment and control groups with similar propensity scores, and then the average difference between them is subsequently estimated. We estimated the propensity score

using a logistic regression model. The explanatory variables include observable characteristics that hypothetically influence the malaria exposure and HRQoL. The characteristics were sex, age, place of residence (urban/rural), socioeconomic status, and dummies for morbidities (diabetes, hypertension, dengue, yellow fever, arthritis, respiratory diseases, and covid). The matching technique used to pair individuals with similar propensity scores was Nearest Neighbor Matching without replacement.

To assess the socioeconomic status of the participant's household, a socioeconomic index was estimated using Multiple Joint Correspondence Analysis (MJCA) according to the presence of household assets. Details of the variables used in the analysis are available in the supplementary material (S1 Fig). The first principal component explained 49.1% of the variability between households. The indicator was normalized from 0 to 100, corresponding to the lowest and highest socioeconomic levels, respectively.

Frequencies were used to characterize the sample and to analyze the components of the EQ-5D instrument. The Pearson chi-square test was used to compare differences between proportions with a significance level of 95% (p < 0.05). Analyses were performed using the R version 4.3.3 statistical software. The packages used for data analysis were included in Tidyverse 2.0.0, MatchIt 4.5.5 for PSM, and ca 0.71_1 for Multiple Joint Correspondence Analysis.

## Results

A total of 1,179 individuals were interviewed. Both sexes were equally represented in the sample (49.96% of female and 50.04% of male). Ages ranged from 18 to 88 years, with a mean and a median of 40 years. The age composition indicates a higher share of younger individuals, with 88.8% aged between 18 and 59 years, and 10.2% aged 60 years or older. Men were in the majority at both the base and the top of the age pyramid (Fig 2). The socioeconomic index had

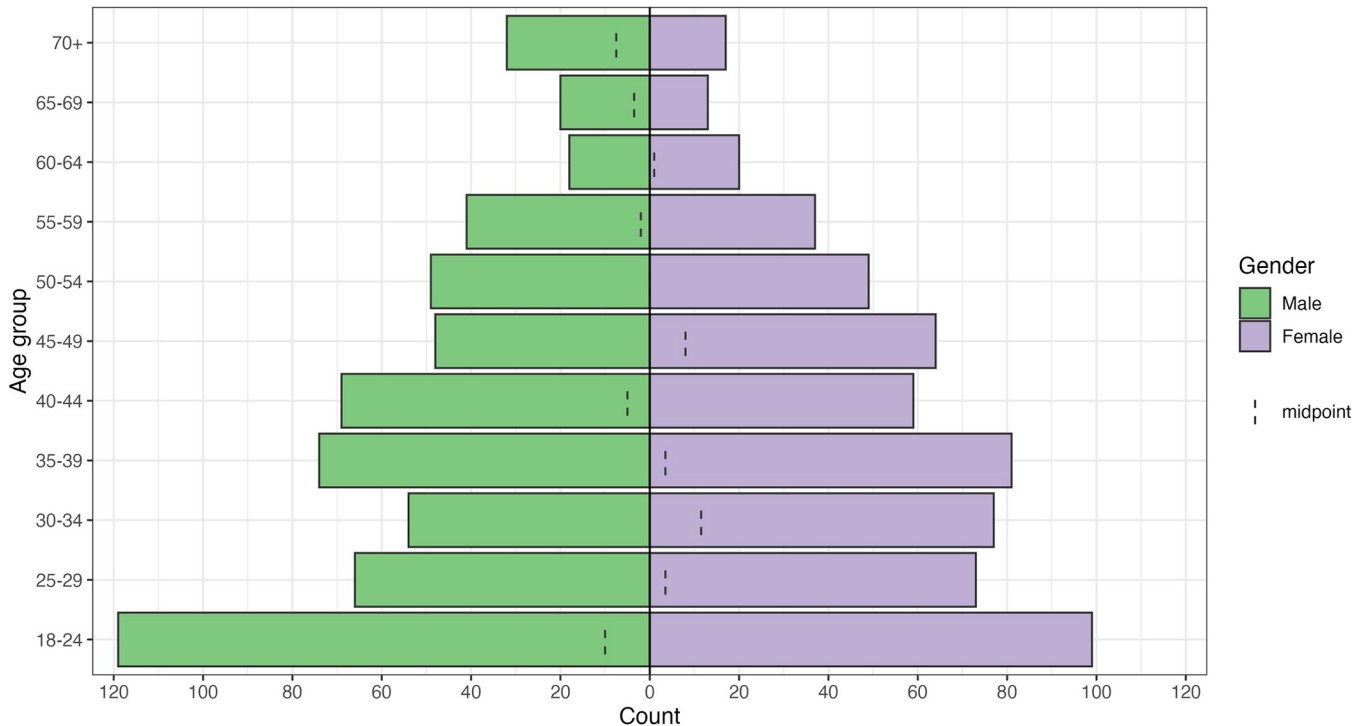

**Fig 2. Demographic distribution of the interviewed individual.**

a mean score of 49.5 on a scale from 0 to 100, indicating that the average socioeconomic status in the sample is near the midpoint of the scale, with a standard deviation of 12.5. In terms of self-perceived health, 9.16% rated their health as very good or good, 35.62% as fair, and 40.88% as poor or very poor. Of those surveyed, 97.3% reported having experienced at least one episode of malaria in their lifetime, with a conditional mean of approximately 11 episodes (median = 6 and mode = 2).

## Matching results

Of the 1,179 individuals interviewed, 500 were in the treatment group, and 679 were in the control group. The baseline characteristics of individuals in the control and treatment groups are described in Table 1. Before matching, the groups differed in terms of sex (P = 0.000), age (P = 0.001), presence of arthritis (P = 0.002), and respiratory problems (P = 0.007). After propensity score matching, the 500 individuals in the treatment group were matched with 500 individuals in the control group, and all observable characteristics were appropriately balanced. Among individuals of the control group, the mean time since the last malaria episode was 1.87 years, with a median of 0.91 years; and approximately 70% of them had their most recent episode more than six months before January 2022.

Fig 3 shows the density of the estimated propensity scores for the treatment and control groups to check for sufficient overlap between the distributions. The graphs show overlap in all propensity score intervals. It was possible to find a match for every individual in the treatment group within the control group in every interval.

## EQ-5D descriptive system and VAS scores

Fig 4 shows the frequencies for each dimension of the EQ-5D-3L descriptive system for individuals in the matched treatment and control groups. Compared to individuals in the treatment group, individuals in the control group reported better health conditions. The percentage of individuals reporting no problems was higher for individuals in the control group in all five health dimensions. Meanwhile, the percentage of individuals reporting moderate or severe problems was higher for individuals in the treatment group in almost all dimensions. "Pain and discomfort" and "anxiety and depression" were the most commonly reported

**Table 1. Sample Characteristics and Covariate Balance in Unmatched and Matched Samples (Means Reported).**

| Variables | Before matching | | After matching | | Treatment |
|---|---|---|---|---|---|
| | **Control** | **P-value** | **Control** | **P-value** | |
| Female | 0.545 | <0.001 | 0.472 | 0.281 | 0.438 |
| Age | 41.689 | <0.001 | 39.968 | 0.187 | 38.728 |
| Socioeconomic Index | 41.092 | 0.057 | 40.249 | 0.473 | 39.683 |
| Urban | 0.300 | 0.390 | 0.296 | 0.339 | 0.324 |
| Covid | 0.361 | 0.809 | 0.354 | 1.000 | 0.354 |
| Diabetes | 0.091 | 0.573 | 0.084 | 0.909 | 0.082 |
| Hypertension | 0.237 | 0.439 | 0.224 | 0.819 | 0.218 |
| Arthritis | 0.144 | 0.002 | 0.080 | 0.649 | 0.088 |
| Breathing problems | 0.135 | 0.007 | 0.080 | 0.731 | 0.086 |
| Dengue fever | 0.066 | 0.068 | 0.072 | 0.731 | 0.096 |
| Yellow fever | 0.046 | 0.067 | 0.028 | 0.846 | 0.026 |
| Number of observations | 679 | | 500 | | 500 |

Note: Treatment is the same before and after matching. P-value is the difference between control and treatment groups.

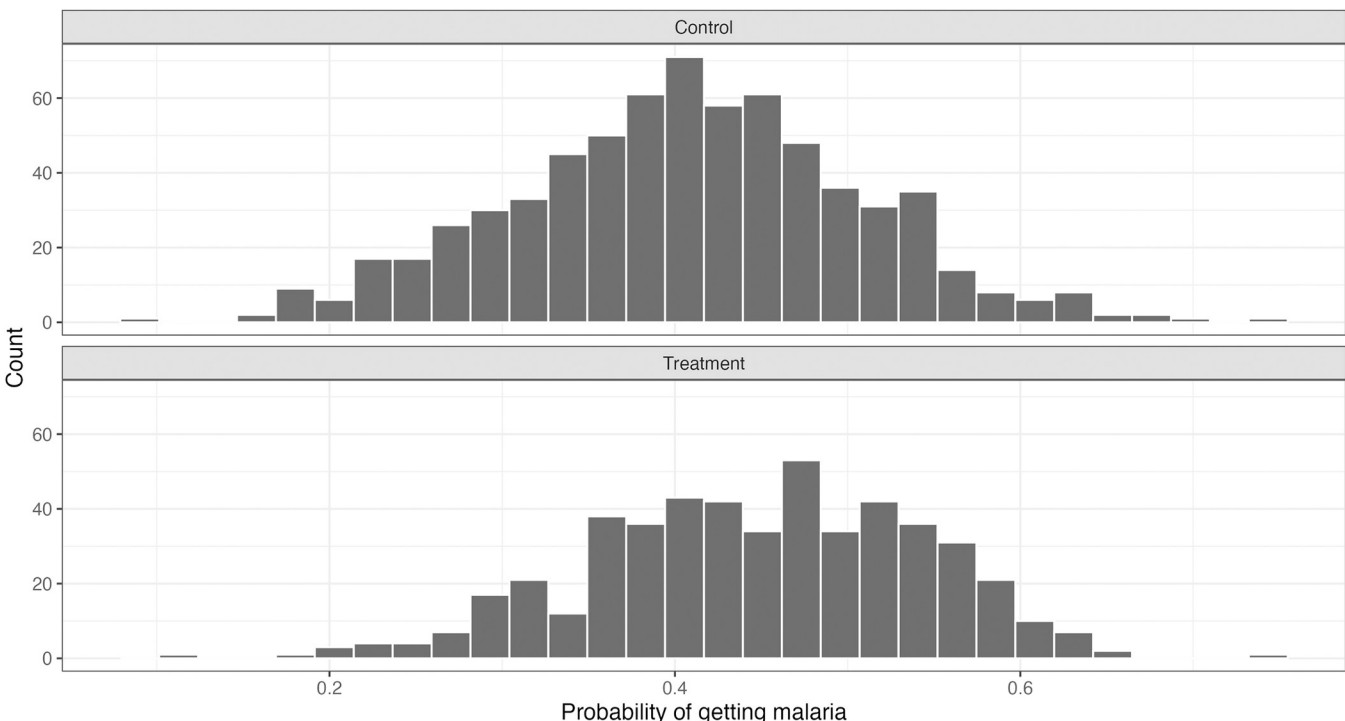

**Fig 3. Histograms of the estimated propensity scores by treatment status.**

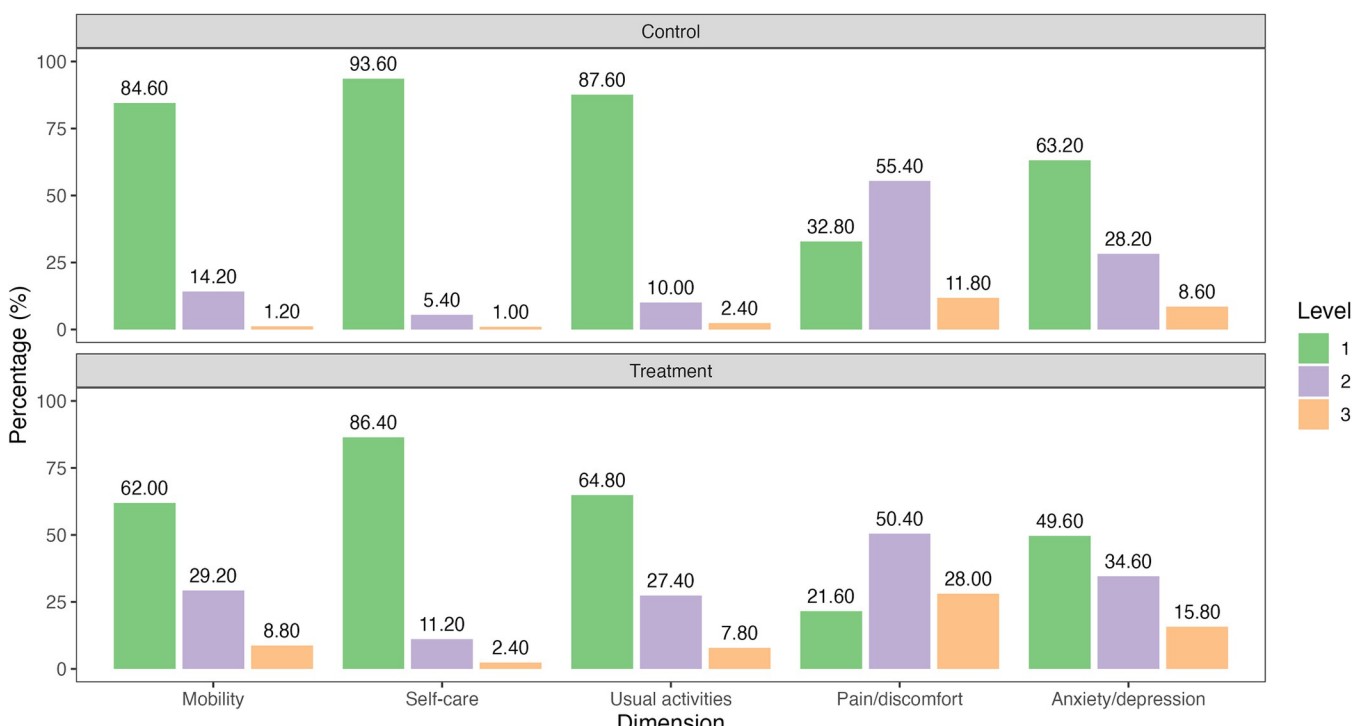

**Fig 4. Distribution of individuals according to the EQ-5D-3L dimension and level of severity for treatment and control groups.**

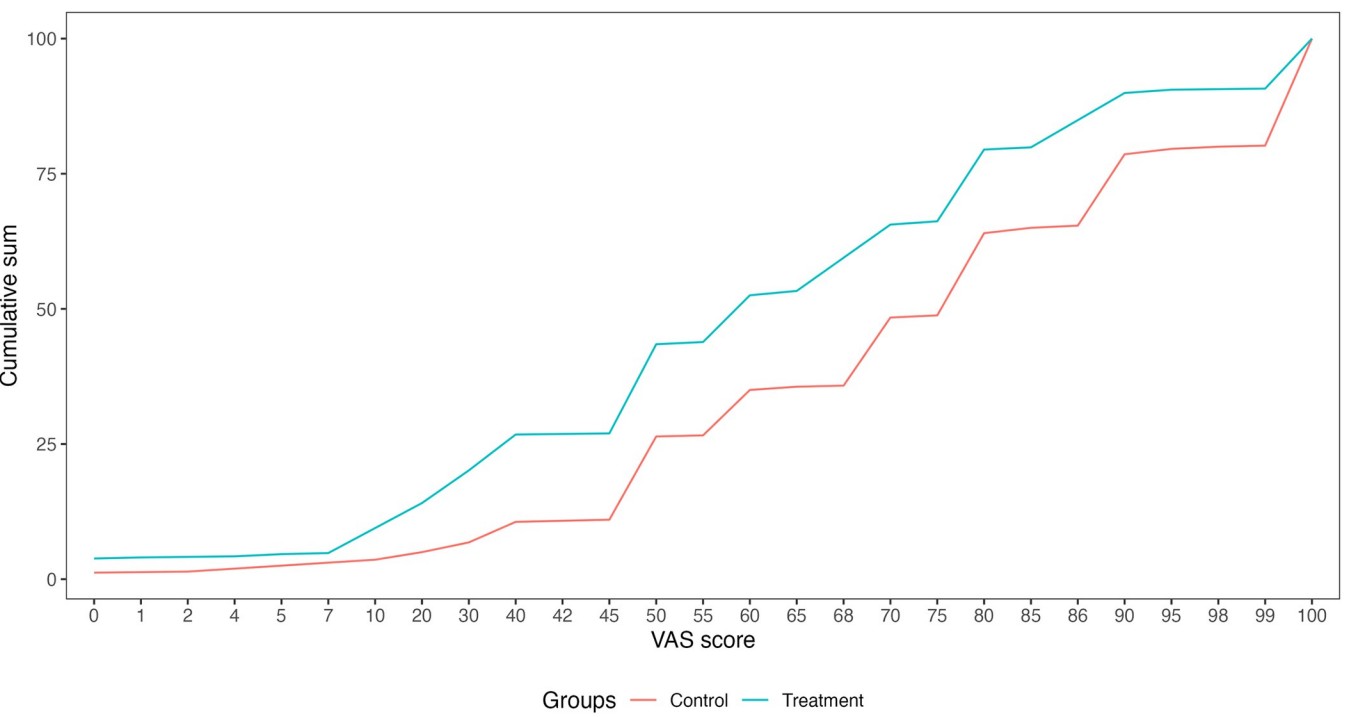

**Fig 5. Cumulative distribution of VAS scores for treatment and control groups.**

problems for individuals in both groups. In the treatment group, 28% of individuals reported severe problems in the "pain and discomfort" dimension, and 15.8% in the "anxiety and depression" dimension. In the control group, 11.8% of individuals reported severe problems in the "pain and discomfort" dimension, and 8.6% in the "anxiety and depression" dimension. The health profiles shown in Fig 4 reveal a population already in poor health states, which are further exacerbated by the presence of malaria.

Fig 5 shows the cumulative distribution of health status scores reported by individuals in the treatment and control groups on the EQ-VAS. The score ranges from 0 to 100, representing the worst and best imaginable health status, respectively. The curve for individuals in the treatment group lies above the curve for individuals in the control group, indicating that the frequency of individuals reporting the worst health states is higher for individuals in the treatment group.

## HRQoL loss due to malaria

The loss of HRQoL due to malaria was estimated by comparing the mean EQ-5D-3L utilities for individuals in the treatment and control groups. A total of 97 health states were identified, with the best health state 11111 occurring 240 times and the worst health state 33333 occurring 4 times (S1 Table). Fig 6 shows the cumulative distribution of health states in descending order of utility for individuals in the treatment and control groups. The mean utility was 0.69 and 0.83 for the treatment and control groups, respectively, representing a loss in HRQoL of approximately 16.3% for individuals experiencing recent malaria episodes. The HRQoL loss measured using EQ-VAS values was quite similar, 17.5% (EQ-VAS value for control group equal to 71.8 versus 59.3 for treatment group).

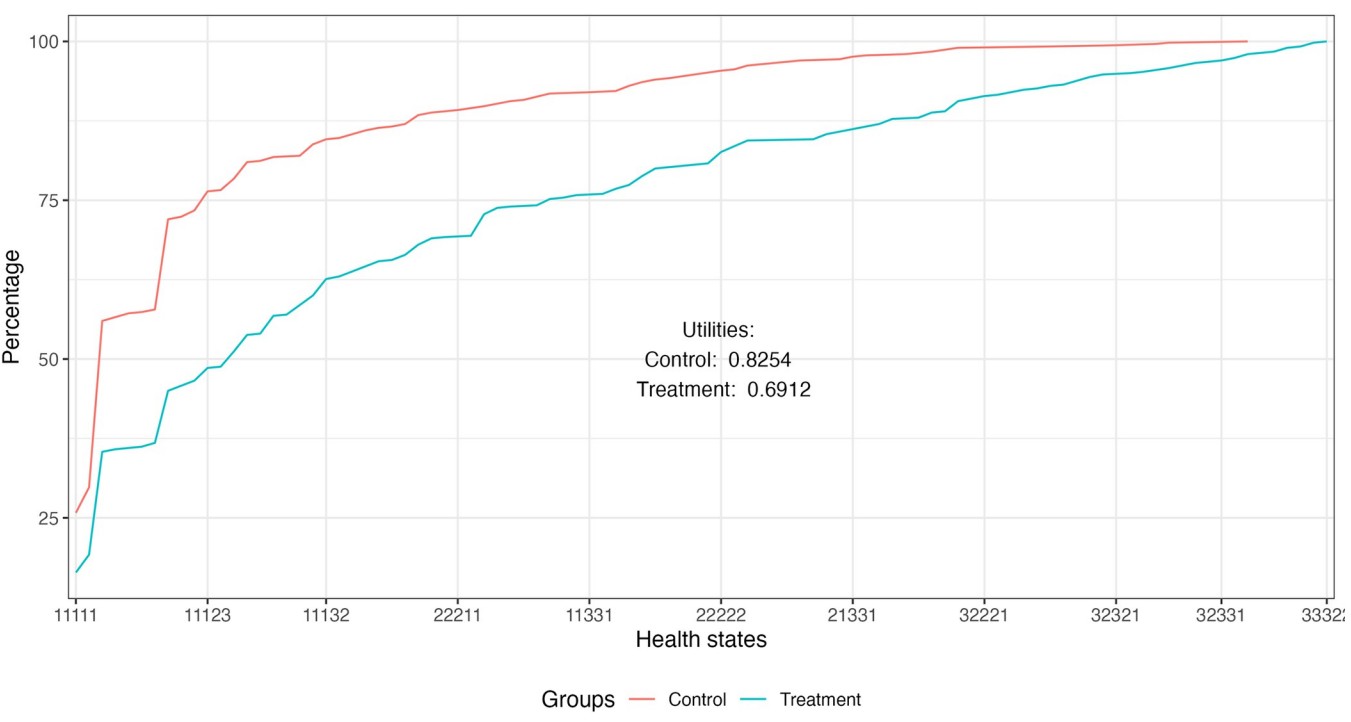

**Fig 6. Cumulative distribution of EQ-5D-3L health states for treatment and control groups.**

## Discussion

This study assessed the loss of HRQoL due to malaria in the population living in endemic areas of the Brazilian Amazon. A total of 1,179 individuals were interviewed in nine municipalities of five states of the region (Roraima, Amazonas, Amapá, Rondônia, and Acre). The mean EQ-5D-3L score was 0.83 for the control group and 0.69 for the treatment group, indicating a loss of approximately 16.3% of individuals with recent malaria episodes. The findings using EQ-VAS values corroborate the EQ-5D-3L results, reflecting a similar magnitude of loss. In addition, the frequency of individuals in the highest severity levels across all five dimensions of the EQ-5D-3L was higher in the treatment group than in the control group. Pain/discomfort was the most commonly reported dimension, followed by anxiety/depression and usual activities. Although not directly comparable, this score is consistent with the EQ-5D-5L scores associated with severe chronic illness. A meta-analysis using a fixed-effects model suggests that HRQoL for malaria episode is similar to that for multiple sclerosis (0.67) and lower than that for chronic kidney disease, chronic obstructive pulmonary disease, and cardiovascular disease (0.76) [16]

HRQoL measures overall health and captures different aspects of an individuals' physical and mental health. Both generic and disease-specific instruments can be used to estimate HRQoL. Generic instruments are typically applied to different populations and health conditions and cover broader dimensions such as physical health, emotional well-being, and social aspects. Disease-specific instruments assess quality of life related to specific interventions, conditions, or groups of patients. They typically focus on specific issues related to a condition, such as symptoms and functional limitations. This study used the generic EQ-5D-3L instrument, developed by the EuroQol Group. EQ-5D-3L is one of the most widely used instruments in cost-effectiveness analysis and it is officially recommended by the National Institute for Health and Care Excellence [17]. It is widely used because of its simplicity, rapid

administration, and broad applicability, allowing comparisons with other groups and interventions. In Brazil, the Ministry of Health guidelines recommend the use of EQ-5D-3L in economic evaluation studies [18]. Since 2013, the EQ-5D social preferences value set has been available for Brazil [14, 15].

Previous studies using the EQ-5D have also found lower levels of HRQoL in patients with malaria episodes, with the most affected domains being pain/discomfort and usual activities. Makatita et al. [10] applied the EQ-5D-5L to a sample of 110 malaria patients from Primary Health Care Centers in Indonesia, finding a mean score of 0.49 for all patients. Scores ranged from 0.349 for those with severe malaria to 0.571 for mild cases. Kayiba et al. [11] used the EQ-5D-3L to estimate the economic burden of uncomplicated malaria in the Democratic Republic of Congo, surveying 1,080 patients and finding a mean EQ-5D index score of 0.62, with the majority reporting moderate or severe problems. Jimam et al. [9] assessed the validity of the EQ-5D-5L to measure HRQoL in cases of uncomplicated malaria in Nigeria and reported good instrument performance with a mean utility score of 0.74. A limitation of these three studies is the lack of control groups, making it difficult to infer HRQoL losses specifically attributable to malaria. To the best of our knowledge this paper is the first to estimate HRQoL losses for a population that has experienced malaria episodes.

The mean EQ-5D score found for the control group in this study (0.83) is lower than previous findings for Brazil: 0.89 for the urban population of Minas Gerais [15] and 0.88 for the metropolitan region of Manaus [19]. In this study, the sampled municipalities are located in endemic areas of the Brazilian Amazon Region, typically characterized by higher socioeconomic vulnerability, difficulty in accessing healthcare services, and poor sanitary conditions. In this context, malaria episodes can exacerbate the vulnerability of health status, in addition to other infectious diseases that are still common in the region, such as yellow fever, hepatitis, dengue, leishmaniasis, leprosy, tuberculosis [20], and intestinal infections [21]. Of those surveyed in this study, 8% reported having had a dengue episode in the previous 12 months, and approximately 4% reported having had yellow fever.

Most respondents had experienced at least one episode of malaria in their lifetime (97.3% of respondents), with 90.24% reporting multiple episodes, yielding a conditional mean of 11. These data underscore the high prevalence of malaria reinfection in the region. Personal prevention measures, such as mosquito nets, repellents, and clothing that protects the arms and legs, can significantly reduce the likelihood of contact with vector mosquitoes [22, 23]. Furthermore, medication adherence reduces the risk of complications and reinfection [24]. However, there is evidence of low individual adherence to preventive measures, such as the use of long-lasting insecticidal nets. Additionally, individuals often interrupt treatment when symptoms improve [24–26]. Adherence to antimalarial treatment is associated with socioeconomic conditions, access to healthcare (access to self-medication, distance to healthcare units, and monitoring of treatment by healthcare agents), knowledge of the disease, and drug administration—treatment duration, presence of symptoms and side effects) [27]. Empirical evidence for the Brazilian Amazon has shown that non-adherence to treatment varies from 30 to 40%, depending on the drug treatment and the method used to measure adherence [28].

The main limitation of this study is the sample design. The field survey was conducted using a convenience sample where municipalities were selected based on the Annual Parasite Index (API) and accessibility conditions. Illegal mining areas and indigenous communities were not included in the sample for security and logistical reasons. As these areas are characterized by the higher levels of malaria incidence and poor socioeconomic conditions and access to healthcare, the HRQoL due to malaria may be underestimated. Another limitation is the retrospective assessment of health status experienced during the last malaria episode. However, as a relatively short interval for recent malaria episodes was considered (up to three

months before questionnaire administration), recall effects may be minimized. Furthermore, the high prevalence and recurrence of malaria in the Amazon region may influence individuals' perceptions of disease risk. Frequent exposure to malaria may lead to familiarity with the disease, resulting in an underestimation of its impact on quality of life.

## Conclusion

To the best of our knowledge, this is the first study to measure malaria-related quality of life loss among the population in endemic areas of the Amazon region. The results show a significant loss of HRQoL due to malaria. These findings underscore the importance of effective malaria prevention and treatment strategies, especially in areas such as the Amazon, where adverse socioeconomic conditions and a challenging epidemiological context exacerbate the impact of the disease.

## Supporting information

**S1 Fig. Box-Plot of items used for the construction of the socioeconomic index using Multiple Joint Correspondence Analysis (MJCA).**
(TIF)

**S1 Table. Health state of HRQOL malaria patients measured with EQ-5D-3L.**
(DOCX)

## Author Contributions

**Conceptualization:** Mônica Viegas Andrade, Kenya Valeria Micaela de Souza Noronha, Gilvan Ramalho Guedes, Nayara Abreu Julião, Lucas Resende de Carvalho, Aline de Souza, Valéria Andrade Silva, Andre Soares Motta-Santos, Henrique Bracarense, Cássio Peterka, Marcia C. Castro.

**Data curation:** Mônica Viegas Andrade, Kenya Valeria Micaela de Souza Noronha, Gilvan Ramalho Guedes, Nayara Abreu Julião, Lucas Resende de Carvalho, Aline de Souza, Valéria Andrade Silva, Andre Soares Motta-Santos, Henrique Bracarense, Marcia C. Castro.

**Formal analysis:** Mônica Viegas Andrade, Kenya Valeria Micaela de Souza Noronha, Gilvan Ramalho Guedes, Nayara Abreu Julião, Lucas Resende de Carvalho, Marcia C. Castro.

**Funding acquisition:** Mônica Viegas Andrade, Marcia C. Castro.

**Investigation:** Mônica Viegas Andrade, Kenya Valeria Micaela de Souza Noronha, Gilvan Ramalho Guedes, Nayara Abreu Julião, Lucas Resende de Carvalho, Aline de Souza, Valéria Andrade Silva, Andre Soares Motta-Santos, Henrique Bracarense, Cássio Peterka, Marcia C. Castro.

**Methodology:** Mônica Viegas Andrade, Kenya Valeria Micaela de Souza Noronha, Nayara Abreu Julião, Lucas Resende de Carvalho.

**Project administration:** Mônica Viegas Andrade, Marcia C. Castro.

**Supervision:** Mônica Viegas Andrade, Marcia C. Castro.

**Validation:** Mônica Viegas Andrade, Kenya Valeria Micaela de Souza Noronha, Gilvan Ramalho Guedes, Nayara Abreu Julião, Lucas Resende de Carvalho, Aline de Souza, Valéria Andrade Silva, Andre Soares Motta-Santos, Henrique Bracarense, Cássio Peterka, Marcia C. Castro.

**Visualization:** Mônica Viegas Andrade, Kenya Valeria Micaela de Souza Noronha, Nayara Abreu Julião, Lucas Resende de Carvalho.

**Writing – original draft:** Mônica Viegas Andrade, Kenya Valeria Micaela de Souza Noronha, Nayara Abreu Julião, Lucas Resende de Carvalho.

**Writing – review & editing:** Mônica Viegas Andrade, Kenya Valeria Micaela de Souza Noronha, Gilvan Ramalho Guedes, Nayara Abreu Julião, Lucas Resende de Carvalho, Aline de Souza, Valéria Andrade Silva, Andre Soares Motta-Santos, Henrique Bracarense, Cássio Peterka, Marcia C. Castro.

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
