## [Decision Letter · Decision Letter 0]

12 Oct 2024

Dear Professor Castro,

Thank you very much for submitting your manuscript "Health-Related Quality of Life due to malaria in the Brazilian Amazon using EQ-5D-3L" for consideration at PLOS Neglected Tropical Diseases. As with all papers reviewed by the journal, your manuscript was reviewed by members of the editorial board and by several independent reviewers. The reviewers appreciated the attention to an important topic. Based on the reviews, we are likely to accept this manuscript for publication, providing that you modify the manuscript according to the review recommendations. 

Sincerely,

Claudia Ida Brodskyn

Section Editor

Claudia Brodskyn

Section Editor

Reviewer's Responses to Questions

**Key Review Criteria Required for Acceptance?**

**Methods**

-Are the objectives of the study clearly articulated with a clear testable hypothesis stated?

-Is the study design appropriate to address the stated objectives?

-Is the population clearly described and appropriate for the hypothesis being tested?

-Is the sample size sufficient to ensure adequate power to address the hypothesis being tested?

-Were correct statistical analysis used to support conclusions?

-Are there concerns about ethical or regulatory requirements being met?

Reviewer #1: I have no specific comments to make about methods other than minor details listed later

Reviewer #2: 1- you definied 3 months as "recent malaria". please explain why it lasts so long

2-please include reference about socioeconomic analysis, why do not use ABEP criteria or other standart SES criteria? please detail the household assets included

**Results**

-Does the analysis presented match the analysis plan?

-Are the results clearly and completely presented?

-Are the figures (Tables, Images) of sufficient quality for clarity?

Reviewer #1: (No Response)

Reviewer #2: 3- Line 184 please include reference for score interpretation

4-Line 208 - VAS instead of EAV

5-Line 220- it is not clear how do you separate the effect of malaria with the previous poor condictions - residual counfounding

6-Line 261 - reference 19 and 20 are incomplete

7- Line 296 it is not prevention, please rewrite the sentence

8- Line 297 I suggest to include the distance from healthcare units, sometimes days

9-Line 300 adhence to treatment or prevention ?

**Conclusions**

-Are the conclusions supported by the data presented?

-Are the limitations of analysis clearly described?

-Do the authors discuss how these data can be helpful to advance our understanding of the topic under study?

-Is public health relevance addressed?

Reviewer #1: (No Response)

Reviewer #2: Please exclude the sentence: "Continued investment in malaria control programs and improved access

 to health services are essential to mitigate the negative impact of this disease on the quality of

 life of affected populations". It is not supported by the data

**Editorial and Data Presentation Modifications?**

Reviewer #1: Points of detail

1. The use of DALY-based estimates of malarial disease burden is mentioned (p5-95). If this is relevant at this point in the text, then it needs some clarification as to the different approach taken with QALY estimates. From this reviewer’s perspective, it might be better located as a point for the discussion.

2. The existence of a 2nd version of EQ-5D is not really relevant at this point and could well be removed (p5-99).

3. (p5-104) EQ-5D is recognised generally as a measure of health-related quality of life (HrQoL) and this term is used later in the manuscript

4. It is tempting to report mean episodes, but it is hard to understand what is meant by 0.73 of an episode (p9-188). Would it be more meaningful to report median or mode as the indicator of central tendency?

5. Table 1 reports sample characteristics, but could benefit from improved labelling, means and percentages? Also, since the treatment group remain unchanged, perhaps the presentation could be rejigged to remove the duplicate?

6. Figure 3 requires a little attention. The 0 / 1 heading is the indicator of group but this needs a text label. Although the x-axis refers to probabilities, the y-axis is labelled “count”. 

7. The statement regarding overlapping propensity score intervals (p11-205) is difficulty to understand from this graphic

8. EAV (p12-208) appears as a section heading, but also in Figure 5. Needs to read VAS?

9. Figure 3 (p13) is actually Figure 4 ?

10. Figure 5 y-axis is cumulative percentage? Would it be clearer to show the distribution along the x-axis and the y-axis to indicate mean EQ-VAS. In any case it would be good to have some gridlines as from inspection it seems that 50% of treatment group report a mean EQ VAS of ~50 whereas the control group is ~70 (almost 50% higher). These data could be compared in the Discussion with the preference-weighted HrQoL scores?

11. Please review the repeated reference to “in selected municipalities of the Brazilian Amazon, 2022” in Figure labels.

12. The Discussion section includes EQ-5D values from studies in other countries, some based on the 5L version. The two versions are based on difference classification/valuation systems, and this distinction must be made here.

Reviewer #2: none

**Summary and General Comments**

Reviewer #1: This study reports the use of the 3L version of EQ-5D in a convenience sample of selected Brazilian municipalities. The study design is straightforward and the use of propensity matching to achieve a balanced control/treatment sample seems perfectly reasonable. 

The results are interesting in that they clearly demonstrate differences between respondents reporting a recent malarial episode and those with a more distal experience (if at all). It would have been interesting to know a little more about the recency of any malaria and/or any residual after-effects in the control group. However, given the results it is fair to conclude (as the authors do) that if anything, the study likely underestimates the true picture.

It would have been interesting also, to see further use of the EQ-5D data, perhaps using the EQ-VAS as the dependent variable to examine covariation of morbidity / study group – for example, was there a larger difference in VAS for respondents with/without a COVID history and was that difference amplified in the control/treatment groups?

More specifically, is there not an obvious further use of the HrQoL results, could they be used to show mean EQ-5D index values by age/sex for the 2 groups? Of course there are limits, but surely it is then a small step to applying those reference values to the municipalities / Region to indicate the potential QALY loss in a single year – even as an order of magnitude? Furthermore, if you monetarise the QALY loss using the World Bank per capita GDP estimate, then this could produce an interesting (even headline-grabbing) result !

Reviewer #2: Original and interesting study

PLOS authors have the option to publish the peer review history of their article (what does this mean?). If published, this will include your full peer review and any attached files.

Reviewer #1: Yes: Professor Paul Kind

Reviewer #2: No

Figure Files:

Data Requirements:

Reproducibility:

References

---

## [Decision Letter · Decision Letter 1]

29 Nov 2024

Dear Professor Castro,

We are pleased to inform you that your manuscript 'Health-Related Quality of Life due to malaria in the Brazilian Amazon using EQ-5D-3L' has been provisionally accepted for publication in PLOS Neglected Tropical Diseases.

Best regards,

Claudia Ida Brodskyn

Section Editor

Claudia Brodskyn

Section Editor

Shaden Kamhawi

co-Editor-in-Chief

Paul Brindley

co-Editor-in-Chief

Reviewer's Responses to Questions

**Key Review Criteria Required for Acceptance?**

**Methods**

-Are the objectives of the study clearly articulated with a clear testable hypothesis stated?

-Is the study design appropriate to address the stated objectives?

-Is the population clearly described and appropriate for the hypothesis being tested?

-Is the sample size sufficient to ensure adequate power to address the hypothesis being tested?

-Were correct statistical analysis used to support conclusions?

-Are there concerns about ethical or regulatory requirements being met?

Reviewer #2: They are all correct

**Results**

-Does the analysis presented match the analysis plan?

-Are the results clearly and completely presented?

-Are the figures (Tables, Images) of sufficient quality for clarity?

Reviewer #2: Correct

**Conclusions**

-Are the conclusions supported by the data presented?

-Are the limitations of analysis clearly described?

-Do the authors discuss how these data can be helpful to advance our understanding of the topic under study?

-Is public health relevance addressed?

Reviewer #2: Yes for all

**Editorial and Data Presentation Modifications?**

Reviewer #2: Accept

**Summary and General Comments**

Reviewer #2: Accept

PLOS authors have the option to publish the peer review history of their article (what does this mean?). If published, this will include your full peer review and any attached files.

Reviewer #2: **Yes: **MARISA SANTOS

---

## [Editor Report · Acceptance letter]

11 Dec 2024

Dear Professor Castro,

We are delighted to inform you that your manuscript, "Health-Related Quality of Life due to malaria in the Brazilian Amazon using EQ-5D-3L," has been formally accepted for publication in PLOS Neglected Tropical Diseases.

Best regards,

Shaden Kamhawi

co-Editor-in-Chief

Paul Brindley

co-Editor-in-Chief
